# Hierarchical Clustering Algorithm for Multi-Camera Vehicle Trajectories Based on Spatio-Temporal Grouping under Intelligent Transportation and Smart City

**DOI:** 10.3390/s23156909

**Published:** 2023-08-03

**Authors:** Wei Wang, Yujia Xie, Luliang Tang

**Affiliations:** 1College of Information Engineering, Nanjing University of Finance & Economics, Nanjing 210023, China; 1120201134@stu.nufe.edu.cn; 2State Key Laboratory of Information Engineering in Surveying, Mapping and Remote Sensing, Wuhan University, Wuhan 430072, China

**Keywords:** intelligent transportation, smart city, computer vision, video GIS, multi-camera, vehicle trajectory, hierarchical clustering

## Abstract

With the emergence of intelligent transportation and smart city system, the issue of how to perform an efficient and reasonable clustering analysis of the mass vehicle trajectories on multi-camera monitoring videos through computer vision has become a significant area of research. The traditional trajectory clustering algorithm does not consider camera position and field of view and neglects the hierarchical relation of the video object motion between the camera and the scenario, leading to poor multi-camera video object trajectory clustering. To address this challenge, this paper proposed a hierarchical clustering algorithm for multi-camera vehicle trajectories based on spatio-temporal grouping. First, we supervised clustered vehicle trajectories in the camera group according to the optimal point correspondence rule for unequal-length trajectories. Then, we extracted the starting and ending points of the video object under each group, hierarchized the trajectory according to the number of cross-camera groups, and supervised clustered the subsegment sets of different hierarchies. This method takes into account the spatial relationship between the camera and video scenario, which is not considered by traditional algorithms. The effectiveness of this approach has been proved through experiments comparing silhouette coefficient and CPU time.

## 1. Introduction

Vehicle trajectory data analysis is a research hotspot in intelligent transportation and smart city [1]. People can deeply understand their life trajectory, social behavior, environmental change, and urban evolution using video GIS, video object recognition, and visual analysis of video object trajectory [2,3]. In recent years, the urban video monitoring system has gradually developed from single-camera processing to multi-camera equipment joint analysis. The system generates a large number of trajectory data under the multi-camera joint monitoring system and records the movement of people, vehicles, and animals in various scenarios [4], with such characteristics as easy deployment, intuitive information, and rich media expression. Dividing each vehicle trajectory into a suitable cluster by measuring its similarity is a crucial and challenging aspect of trajectory analysis [5]. However, there are some areas for improvement in current clustering methods for video object trajectory research. Specifically, the analysis object is limited to image trajectory [6], ignoring the actual trajectory of the video object in geographical space. Additionally, the traditional trajectory clustering algorithm does not consider the geographic space relation of multi-cameras and only clusters datasets in a single camera and a single scenario, resulting in poor real-time performance and accuracy in large-scale cross-camera or even cross-camera group trajectory data.

Clustering large-scale trajectory data is challenging. First, the trajectory is complex, so the computational cost for clustering analysis is high. Second, the trajectories sampled by the multi-camera joint monitoring system are not equidistant, making it difficult to unify the dimensions between different video object trajectories, so reasonable point correspondence rules should be given to measure the difference in distance calculation. Finally, there is a vision-blind area and a long distance between camera groups, so the direct calculation without simplifying trajectories causes the local differences between trajectories to be overlooked due to the large global scope. The traditional algorithms only cluster the equidistant trajectories of sampling points under a single camera for small-scale datasets without considering the hierarchical relation between camera position, the field of view, and the motion of video objects between cameras and scenarios and without reasonable analysis of cross-group trajectory data. Therefore, this paper proposed a hierarchical clustering algorithm for multi-camera vehicle trajectories based on spatio-temporal grouping. Based on our previous work, we have done further research in this paper [7]. In this paper, the camera groups are divided according to the spatial distribution law of camera equipment in reality, which effectively makes up for the deficiency of traditional methods that do not take into account the spatial relationship between camera equipment. The point correspondence rule of optimal unequal length trajectories and the overlapping scale factor of trajectory distance under camera-joint system are proposed to effectively calculate the distance between unequal multi-camera trajectories and effectively reduce the computational cost compared with the traditional methods. In addition, this paper also proposes a method to automatically obtain labels of video object trajectories in the camera group and across camera groups, which can achieve better clustering results through additional useful supervision information.

This paper is mainly structured as follows: Section 2 introduces the relevant work in the research field involved; Section 3 presents the method description, including the spatialization of video object trajectory, scenarios and division rule of camera groups, point correspondence rule of optimal unequal length trajectory, overlapping scale factor of trajectory distance under camera-joint system, and the details of the supervised video object clustering algorithm within and between camera groups; Section 4 demonstrates the visualization effect of the algorithm on the dataset and evaluates the accuracy and efficiency of the method through experiments; Section 5 discusses the implications of this work, as well as the limitations of this work and how it will be addressed in future works; Section 6 presents the conclusions and prospects.

## 2. Related Work

The research fields involved in this paper include video-geographic scenario data fusion organization and trajectory clustering. The research status of relevant fields is as follows.

### 2.1. Video-Geographic Scenario Data Fusion Organization

The data fusion organization of video-geographic scenarios is the basis of video object analysis combining geospatial information. Based on the concepts of Multi Media GIS [8], Geo Video [9], and Video GIS [10], the metadata description method [11] and GPS correlation method [12] were constructed in the earlier research, and geographic retrieval and play of video images were realized by describing the geographic location of video frames.

In recent years, research has focused more on the fusion of video content and geographic scenarios [13]. The fusion of video content and geographic scene can be subdivided into the method of strengthening video through geographic scene and the method of strengthening geographic scene by video [14]. The output result of the former is enhanced video, and the video content is visually enhanced by using the spatial information in the geographic scene [15], while the latter is constructed and presented through the geographic scene [16]. According to the different mapping objects, it can be divided into video frame picture projection and video object projection. The video frame picture projection maps the video frame image to the geographic scene according to the camera parameters. The video object projection realizes the separation of the front and rear scenes through the related theory of computer vision and projects into the geographical scene. The picture projection of video frame includes the methods of correlation [17], fixed plane play [18], global mapping [19], and so on. The video object projection can be divided into the methods of foreground and background independent projection [20], foreground projection [21], and foreground abstraction [22].

### 2.2. Trajectory Clustering

Common in pattern recognition, data analysis, and machine learning, trajectory clustering is an efficient method for analyzing trajectory data, aiming to obtain space, time, and even potential information in trajectory data. It is widely applied in such fields as object motion prediction [23], traffic monitoring [24], behavior understanding [25], anomaly detection [26], and 3D reconstruction [27]. In addition, data representation, feature extraction, and distance measurement selection are the key preliminary work of trajectory clustering. For example, a trajectory can be represented as a vector and downsampled to a uniform length, so the Euclidean distance [28] can be used. Trajectories can also be considered as samples of the probability distribution, so Bhattacharyya distance [29] can be used to measure the distance between two trajectories.

According to the availability of labeled data, there are three trajectory clustering methods: unsupervised, supervised, and semi-supervised models. The unsupervised model aims to cluster the data without human experts supervising or labeling the data and obtain the reasoning function by analyzing the unlabeled dataset [30]. The supervised model is learned before trajectory clustering. The labeled data are usually used to learn the function that maps data to its label, the function that predicts unlabeled data clustering [31]. The labeled data often require heavy manual work by experts. The semi-supervised algorithm is trained by labeled data and adjusted by unlabeled data [32].

The unsupervised trajectory clustering algorithm does not rely on other prior knowledge but uses constructors to describe the implicit relationship between trajectory samples. The representative methods are densely clustering models [33], hierarchical clustering models [34], and spectral clustering models [35]. By directly defining the density-reachable rule, densely clustering models can obtain the maximum set of density connected samples as cluster clusters. Hierarchical clustering models are divided into “top-down [36]” type and “bottom-up [37]” type. The former regards each trajectory sample as a cluster and gradually merges into larger clusters by defining the similarity between clusters; the latter gradually divides the trajectory data set into smaller groups by using the idea of divide and conquer until it meets the requirements of clustering. Supervised trajectory clustering algorithm is represented by the K-NN algorithm [38] and statistical model method [39]. The K-NN algorithm uses the average nearest distance as the evaluation criterion to realize trajectory clustering. The statistical model method uses the Gaussian mixture model [40] and other statistical models to construct the trajectory category probability function to obtain the trajectory sample category. In addition to the above methods, in recent years, with the continuous extension of deep learning, there are a large number of deep learning algorithms [41,42]. Semi-supervised trajectory clustering algorithm [43,44] fully combines supervised clustering algorithm with unsupervised clustering algorithm to reduce the label labor cost and minimize the over-fitting problem in the clustering process. The basic idea is to update the classifier by classifying the trajectory data and clustering the new data.

In order to understand the three kinds of algorithms more clearly, we give the tabular overview of different trajectory clustering algorithms in Table 1.

## 3. Method Description

### 3.1. Spatialization of Video Object Trajectory

Building the mapping relation between each camera image square plane and geographical space object square plane is necessary to transform from image space coordinates to geographical space coordinates. This paper used the contact point between the video object subgraph and the ground as the locating point and sampled at a certain time interval [45]. The video object trajectory in the geographic scenario was obtained from a building mapping relation between image space and geographic space [46], as shown in Figure 1.

This study uses the homography matrix method to construct a mapping model. If the image coordinate of a point is q, and the geographic space coordinate is Q, then the homogeneous coordinate of q and Q can be represented as
(1)$$q={\left[x\;\;y\;\;1\right]}^{T},$$
(2)$$Q={\left[X\;\;Y\;\;Z\;\;1\right]}^{T}.$$

If the mapping matrix is M, the relational expression between q and Q is
(3)q=MQ.

The image plane is scaled, translated, and rotated to a geospatial plane, so the mapping matrix M can be decomposed into
(4)M=s·W·R,
where s is the scale factor; W is the camera translation transformation matrix; R is a 3 × 4 dimensional rotation transformation matrix.
(5)$$W=\left[\begin{array}{ccc} f_{u} & 0 & u\\ 0 & f_{v} & v\\ 0 & 0 & 1 \end{array}\right]$$
(6)$$R=\left[r_{1}\;\;r_{2}\;\;r_{3}\;\;e\right]$$
where $f_{u}$ and $f_{v}$ represent the product of the physical focal length of the lens and the sensor size in the horizontal and vertical axis directions of each unit, respectively; u and v represent the offset of the imaging center relative to the principal optic axis on the horizontal and vertical axis, respectively; $r_{1}$, $r_{2}$, and $r_{3}$ represent the rotation relation of the coordinate system in the X-axis, Y-axis, and Z-axis in the physical space, respectively; e is the translation relation between coordinate systems.

When using the homography matrix method, it is assumed that the camera field of view plane in geographic space is horizontal—that is, Z = 0 at the plane. Thus, the mapping relation between image and geographic space can be regarded as the mapping from one plane to another. To simplify the calculation, the Z in Q and the $r_{3}$ rotated about the Z-axis in R are removed. Hence, the homography matrix M is simplified as
(7)$$M=s\cdot \left[\begin{array}{ccc} f_{u} & 0 & u\\ 0 & f_{v} & v\\ 0 & 0 & 1 \end{array}\right]\cdot [\;r_{1}\;\;\;r_{2}\;\;\;r_{3}\;\;e\;].$$

The geospatial coordinates of the video object trajectory can be obtained according to the solution of the matrix M.

### 3.2. Scenarios and Division Rule of Camera Groups

The closely adjacent location cameras within the group and the overlapped camera field of view are preferred when cameras are divided into groups. The weigh options rules between the two can be specified as
(8)$$Sel_{rule}=\frac{1}{\varepsilon }\underset{\mathrm{cam}\in C}{\mathrm{argmax}}P\left(D|h\right)P\left(h\right),$$
where C represents all cameras to be divided, and P(h) represents the nearest neighbor in a position, which can be stated as the reciprocal of the distance between positions:(9)$$P\left(h\right)=\frac{1}{\sum \left\|loc\left({cam}_{u}\right)-loc\left({cam}_{v}\right)\right\|}.$$

P(D|h) is the degree of view overlap, expressed by the area of view overlap:(10)$$P\left(D|h\right)=\sum area({cam}_{u})\cup area\left({cam}_{v}\right).$$

ε represents the normalized constant parameter.

### 3.3. Point Correspondence Rule of Optimal Unequal Length Trajectories

Inspired by the dynamic time warping [47], to better compare the distances between two trajectories with different lengths, it is necessary to establish one-to-many and many-to-one matching so that the two trajectories have the same pattern of wave troughs and peaks perfectly matched. Two unequal-length trajectory sequences are known:(11)$${Tra}_{x}=\left\{p_{x,j}\left(j=1,\cdots ,\;{len}_{x}\right)\right\},{\;Tra}_{y}=\left\{p_{y,j}\left(j=1,\cdots ,\;{len}_{y}\right)\right\},\;{\;len}_{x}\neq {len}_{y}.$$

The trajectory sequence may not have equidistant time points. A feature space represented by F is fixed. Its local distance measure is defined as a function:(12)c:F×F→R≥0.

The defined sequence is the point correspondence sequence of optimal unequal-length trajectory:(13)$$Seq\left(S\right)=\left(s_{1},s_{2},\cdots ,s_{e},\cdots ,s_{len\left(Seq\left(S\right)\right)}\right).$$
(14)$$s_{e}=\left(n_{e},m_{e}\right)\in \left[1:N\right]\times \left[1:M\right]$$

Then, the sequence $Seq\left(S\right)$ satisfies
(15)$$\left\{\begin{array}{c} s_{1}=(1,1)\\ s_{len(Seq\left(S\right))}=(N,M)\\ \begin{array}{c} n_{1}\leq n_{2}\leq ...\leq n_{len\left(Seq\left(S\right)\right)},m_{1}\leq m_{2}\leq ...\leq m_{len\left(Seq\left(S\right)\right)}\\ s_{e+1}-s_{e}\in \{(1,0),(0,1),(1,1)\} \end{array} \end{array}\right..$$

The distance cost between ${Tra}_{x}$ and ${Tra}_{y}$ is written as
(16)$$c_{p}\left(\;{Tra}_{x},{Tra}_{y}\;\right)=\sum_{l=1}^{L}c({{Tra}_{x}}_{n_{l}},{{Tra}_{y}}_{m_{l}}).$$

The optimal total cost between ${Tra}_{x}$ and ${Tra}_{y}$ is shown as the minimum value of $c_{p}\left({Tra}_{x},{Tra}_{y}\right)$:(17)$$\min\left\{c_{p}\left(\;{Tra}_{x},\;{Tra}_{y}\;\right)\right\}.$$

### 3.4. Overlapping Scale Factor of Trajectory Distance under Camera-Joint System

This part explains the increase in distance calculation due to the time overlap of multiple cameras in the camera-joint system. Then it gives a method to obtain the scale factor of trajectory distance overlap to offset the increase in distance.

Assuming G as a camera group,
(18)$$G=\{{Cam}_{1},{Cam}_{2},{Cam}_{3},\ldots \ldots ,{Cam}_{n}\}.$$

For ∀ video object ${Obj}_{i}\in G$, it is assumed that ${Obj}_{i}$ is captured by $G_{i}=\left\{{Cam}^{\left(1\right)},{Cam}^{\left(2\right)},\ldots ,{Cam}^{\left(s\right)},\ldots ,{Cam}^{\left(t\right)}\right\}$, where
(19)$$G_{i}\subseteq \;G.$$

According to the capture sequence, the trajectory corresponding to the video object can be expressed as a series of nodes:(20)$${Tra}_{i}=\left\{p_{i,j},\left(j=1,\ldots ,{len}_{i}\right)\right\},$$
(21)$$p_{i,j}=\left({Cam}_{i,j},x_{i,j},y_{i,j},t_{i,j}\right),$$
where
(22)$${Cam}_{i,j}\in G_{i},$$
which represents that the video object is captured by ${Cam}_{i,j}$ at this node.

Assuming the two trajectories ${Tra}_{1}$ and ${Tra}_{2}$ under the camera group formed by cameras ${Cam}_{1}$ and ${Cam}_{2}$ are shown in Figure 2.

In ${Tra}_{1}$, the total duration under two cameras is $t_{1}$, which can be divided into $t_{2},{\;t}_{3},\;\;t_{4}$, in which $t_{3},\;\;t_{4}$ are captured by only one camera while $t_{2}$ is captured by two cameras at the same time; ${Tra}_{1}$ is only captured by ${Cam}_{1}$ during time $t_{5}$. If the distance between the two is calculated directly, the density of trajectory nodes will increase as cameras overlap in time $t_{2}$ of ${Tra}_{1}$, leading to an increase in the distance value. Therefore, ${Tra}_{1}$ is obtained first to eliminate the distance of the added part. The total global duration of ${Tra}_{2}$ is $\alpha_{1}$ = $t_{1}$, $\alpha_{2}$ = $t_{5}$, respectively, while the actual calculation duration is $\beta_{1}$ = $t_{1}$ + $t_{2}$, $\beta_{2}$ = $t_{5}$. Therefore, the scale factor of overlapping trajectories of ${Tra}_{1}$ and ${Tra}_{2}$ can be obtained as follows:(23)$$k_{i}=\frac{\alpha_{i1}+\alpha_{i2}}{\beta_{i1}+\beta_{i2}}.$$

This factor should be multiplied in the distance calculation to eliminate the increase in distance due to camera overlap.

This part details the trajectory clustering algorithm within the camera group (SCAIMG) and between camera groups (SCAIBG). The process of the algorithm is shown in Figure 3.

#### 3.4.1. Supervised Clustering Label Acquisition

Supervised clustering results can be obtained with more additional useful supervised information than unsupervised clustering. For the video object trajectory under the camera joint system in this chapter, the “main camera” of video objects is defined as the label for the supervised clustering of trajectory samples. For the trajectory between camera groups, the number sequence of the camera group passed through is used as the label of the supervised clustering of the trajectory sample.

In the camera group of the camera-joint system, the camera number with the most capture times should be the same among video objects with the same characteristics. Therefore, for accurate trajectory data clustering, we use the camera number with the highest capture times corresponding to each video object as the label of supervised clustering:(24)$$label\left({Tra}_{i}\right)=\underset{{\mathrm{Cam}}_{i,j}\;\mathrm{correspond}\;\mathrm{to}\;{\mathrm{Tra}}_{i}}{\max}{NUM(Cam}_{i,j}),$$

Furthermore, we used the camera group number as the label of supervised clustering between camera groups.

#### 3.4.2. Trajectory Clustering Algorithm within the Camera Group

The trajectory clustering algorithm within the camera group aims to obtain a group of clustering centers. First, the video objects in the camera group are spatialized, followed by clustering their trajectory. The pseudo-code of trajectory clustering within the group (SCAIMG) is shown in Algorithm 1.
**Algorithm 1:** Supervised trajectory clustering algorithm considering camera information in the multi-camera collaborative monitoring group (SCAIMG)$1:\;\mathrm{Input}:\;A\;\mathrm{camera}\;\mathrm{group}\;G=\{{Cam}_{1},{Cam}_{2},{Cam}_{3},\ldots \ldots ,{Cam}_{n}\}$$,\;\mathrm{the}\;\mathrm{trajectory}\;\mathrm{set}:\;T_{G}=\left\{{Tra}_{i}\right\},{Tra}_{i}=\left\{p_{i,j},\left(j=1,\ldots ,{len}_{i}\right)\right\}$.$2:\;\mathrm{Output}:\;a\;\mathrm{group}\;\mathrm{of}\;\mathrm{trajectory}\;\mathrm{clustering}\;\mathrm{centers}\;C=\{c_{1},c_{2},\ldots c_{\lambda }\}$.3:  Obtain the capture time of each video object under each camera according to Section 4.1$\;\mathrm{and}\;\mathrm{use}\;\mathrm{the}\;\mathrm{camera}\;\mathrm{number}\;\mathrm{with}\;\mathrm{the}\;\mathrm{most}\;\mathrm{capture}\;\mathrm{times}\;\mathrm{as}\;\mathrm{the}\;\mathrm{label}:\;label\left({Tra}_{i}\right)=\underset{{\mathrm{Cam}}_{i,j}\in {\mathrm{Tra}}_{i}}{\max}{NUM(Cam}_{i,j})$$.\;\mathrm{Initialize}\;\mathrm{the}\;\mathrm{new}\;\mathrm{set}\;\Phi =T_{G}$, set clusters number K$,\;\mathrm{initialize}\;\mathrm{set}\;all\_vectors\_updated=\left[0,0,\ldots ,0\right](size=K)$; Randomly select K samples from Φ$\;\mathrm{as}\;\mathrm{initial}\;\mathrm{cluster}\;\mathrm{centers}\;C\leftarrow \{c_{1},c_{2},\ldots c_{\lambda }\}$; Set the number of iterations parameter epochs$;\;\mathrm{Set}\;\mathrm{the}\;\mathrm{learning}\;\mathrm{rate}\;\eta \in \left(0,1\right)$
$4:\;\mathrm{According}\;\mathrm{to}\;\mathrm{Equation}\;(23),\;\mathrm{obtain}\;\mathrm{the}\;\mathrm{overlap}\;\mathrm{scale}\;\mathrm{factor}\;\mathrm{of}\;\mathrm{each}\;\mathrm{trajectory}:\;k_{i}\leftarrow \frac{\alpha_{i1}+\alpha_{i2}}{\beta_{i1}+\beta_{i2}}$.5: step=06: While not each vector in C is updated and step < epochs:7:   For each sample s from Φ:8:       calculate the distance between s$\;\mathrm{to}\;\mathrm{each}\;\mathrm{center}\;c_{\tau }$$:\;d\left(s,c_{\tau }\right)\leftarrow min\left\{c_{p}\left(s,c_{\tau }\right)\right\}$9:      cτ*←cτ,where τ*=arg minτ∈1,2,…,K⁡ds,cτ$10:\;\;\;\;\;\mathrm{If}\;label\left(s\right)==label\left(c_{\tau }\right):$$11:\;\;\;\;\;\;\;s^{\prime }\leftarrow s/(1-\eta )\cdot d\left(s,c_{\tau }\right)$12:     Else:$13:\;\;\;\;\;\;\;s^{\prime }\leftarrow s/(1+\eta )\cdot d\left(s,c_{\tau }\right)$$14:\;\;\;all\_vectors\_updated\left[s\right]=1$15: Return Φ

In Algorithm 1, line 11 and line 13 represent the operations of “approaching” and “moving away”, respectively. The schematic diagram of the approaching operation is shown in Figure 4. It is assumed that $label\left(s\right)==label\left(c_{\tau }\right)$ is known, and every node in the trajectory s moves towards $c_{\tau }$.

The trajectory of the “moving away” operation moves in the opposite direction in Figure 4.

#### 3.4.3. Trajectory Clustering Algorithm between Groups

We can combine the entry and departure points in each camera field of view of video object trajectories to reflect the dynamic characteristics of group trajectories [48,49] because the range of trajectories between groups is larger than that within a group. The entry point and departure point of the vehicle video object under each camera group are taken as the trajectory sampling points to form multiple trajectory subsegments, which are then hierarchically clustered.

The sampled trajectory is supposed as:(25)$$f_{i}=\left\{p_{i,j},\left(j=1,\ldots ,{len}_{f_{i}}\right)\right\},$$
(26)$$p_{i,j}=\left({Group}_{i,j},x_{i,j},y_{i,j},t_{i,j}\right),$$
where $p_{i,j}$ represents the entry point or departure point information, and ${Group}_{i,j}$ represents the j-th camera group number that video $f_{i}$ passes through. $x_{i,j},y_{i,j}$ represent the coordinates, and $t_{i,j}$ represents the timestamp.

Figure 5 shows the trajectory $f_{1},f_{2},f_{3}$ in camera Groups 1–3. $f_{1}$ passes through Group 1, Group 2, and Group 3, with five hierarchies; $f_{2}$ passes through Group 1 and Group 2, with three hierarchies; $f_{3}$ passes through Group 3, with one hierarchy. $f_{1},f_{2}$ contain trajectory subsegments of three hierarchies starting at Group 1 and ending at Group 2; $f_{1},f_{3}$ contain trajectory subsegments in Group 3, with one hierarcy.

We proposed a trajectory clustering algorithm between camera groups with multi-hierarchies to reasonably analyze the cross-camera group trajectories. The algorithm aims to obtain the λ sets of cluster centers:(27)$$\left\{C\left(1\right),C\left(3\right),\ldots C\left(2h-1\right),\ldots C\left(\lambda \right)\right\},h\in N,$$
where
(28)$$C\left(2h-1\right)=\left\{c_{2h-1,1},c_{2h-1,2}\ldots ,c_{2h-1,k},{\ldots ,c}_{2h-1,l}\right\},\;h,k\in N,$$
(29)$$c_{2h-1,k}=average\left(\left\{f_{2h-1,m},f_{2h-1,m+1},\ldots ,f_{2h-1,m+2\left(\lambda -1\right)}\right\}\right),$$
(30)$$\left\{f_{2h-1,m},f_{2h-1,m+1},\ldots ,f_{2h-1,m+2\left(\lambda -1\right)}\right\}\in c_{2h-1,k},$$
where λ refers to the number of cross-camera groups of trajectories within a group, $C\left(2h-1\right)$ refers to the cluster center set when the number of cross-camera groups of trajectories within groups is 2h−1, and k refers to the k-th cluster center.

The pseudo-code of Supervised trajectory clustering algorithm considering camera information between groups (SCAIBG) is shown in Algorithm 2.
**Algorithm 2:** Supervised trajectory clustering algorithm considering camera information between groups (SCAIBG)1: Input: a group cross-camera groups video object trajectories: fi=pi,j,j=1,…,lenfi2: Output: λ groups of trajectory clustering centers {C1,C2,⋯,Cλ}.3: For t ← 1 to λ:4:  Initializeanewset Φ5:    For u ← 1 to obj_num:6:      If Cnumfobju≥ λ://Cnumfobju  indicates the total number of camera groups passed by obju7:       For v ← 0 to u−λ−1:8:         Add fuv, fuv+1,…, fuv+λto Φ//fuvrepresents the v-th trajectorysubsegment of the u-th videoobject9:  Set clusters number Kλ10:  Randomly select Kλ samplesfrom Φ as Initial cluster centers:Cλ=ct,1,ct,2,…,ct,Kλ 11:  Set epochs as the maximum number of iterations12:  Repeat:13:    For each sample s from Φ:14:      ct,w*←ct,w, where w*=arg minw∈1,2,…,Kλds,ct,w15:      If label(s) = label(ct,w*):16:       s′=s/1−η·ds,ct,w*17:      Else18:       s′=s/1+η·ds,ct,w*19:  Until iterations exceed epochs or there is no change in cluster centers20:  t ← t+121: ReturnΦ


## 4. Experimental Analysis

This part briefly introduces the experimental conditions and data and the trajectory clustering results within and between camera groups. Compared with traditional clustering algorithms, the advantages of the proposed algorithm are demonstrated by silhouette coefficient evaluation, and the time complexity is analyzed to prove the algorithm’s effectiveness.

### 4.1. Experimental Conditions and Data

The experimental data in this paper come from the CityFlowV2 dataset [50] newly launched by NVIDIA, the first large-scale dataset in the world to support cross-camera vehicle tracking, accommodating more than ten intersections and 40 cameras at the same time, with a spatial span greater than 3 km^2^. This dataset contains high-definition synchronous videos collected in Dubuque, the United States, including scenarios such as residential areas and expressways.

The experimental data in this paper are all the original dataset’s video sequences. The experimental environment comprises software (Windows10, python 3.6 + sklearn0.0) and hardware (Intel (R) Core (TM) i7-10510U CPU @ 1.80 GHz 2.30 GHz, RAM 12.0 GB, and NVIDIA GeForce MX250). The algorithm to obtain the video object trajectory by preprocessing is as follows: the video dynamic object detection algorithm is Mask-RCNN [51], the tracking algorithm is TNT [52], and the cross-camera re-recognition algorithm is Deep SORT [53].

### 4.2. Camera Grouping of the CityFlowV2 Dataset

The CityFlowV2 dataset is divided into groups according to Equations (9) and (10). Partial division results are shown in Figure 6. In addition to the groups shown in Figure 6, Cam 1 to Cam 5 are a group, and Cam 6 to Cam 9 are a group.

### 4.3. Determination of the Number of Cluster Centers

The number of cluster centers is determined using the elbow method [54]. Due to the continuous optimization of the cluster center and smaller quadratic sum, the first inflection point appearing in the change of the quadratic sum is chosen as the best K value.
(31)$$min\left(SSE\right).$$
(32)$$SSE=\sum_{k}^{i=1}\underset{p\in C_{i}}{\sum }{\left|p-m_{i}\right|}^{2}.$$

For SCAIMG and SCAIBG, the cluster center value of each camera group is determined using the elbow method, as shown in Figure 7 and Figure 8. The number of cluster centers of each camera is the abscissa K corresponding to the blue box.

According to Figure 7 and Figure 8, we can respectively determine the reasonable number of cluster centers K according to the number of cluster centers in each group—the inflection point in the curve of average distortion degree.

### 4.4. Determination of the Initial Cluster Center

If the initial cluster centers for SCAIMG and SCAIBG are randomly selected, this usually results in slightly different clusters at the end when the clustering algorithm is rerun-ed. In order to solve the above problem, we refer to the work of the other literature [55], make improvements, and propose the initial cluster center acquisition method.

Assume that there are L. trajectories for clustering, and the number of supervised clustering label categories is ζ. First, a trajectory sample is selected from each category—that is, a total of ζ trajectory samples are selected as the initial cluster centers, and then the sample with the largest average distance from ζ. centers is selected from the remaining L−ζ samples as the ζ+1-th initial cluster center, and so on until K initial cluster centers are selected.

### 4.5. Algorithm Results and Effect Evaluation

#### 4.5.1. Trajectory Clustering Results within a Group

First, we demonstrate the visualization results of clustering ten camera groups with overlapping cameras in each group. For example, the visualization effect of the trajectories in group 1 is shown in Figure 9:

The trajectories under each camera group were obtained in chronological order. The clustering results of SCAIMG in each group are shown in Figure 10. The parameters are set to epochs=500,η=0.01. The red line represents the vehicle trajectory, while the green line is the cluster center.

Through the visualization effect, a small number of cluster centers represent the overall trend of trajectory data of 10 camera groups.

#### 4.5.2. Trajectory Clustering Results between Camera Groups

In this paper, we performed clustering analysis on trajectory subsegments with three and five hierarchies. There are trajectory subsegments with seven, nine, and eleven or even more hierarchies. However, in these cases, there is a small number of trajectory subsegments, making the clustering analysis lose its practical significance because clustering is to extract a large number of trajectories of the overall trend.

The number of the trajectory subsegments at different hierarchies is shown in Table 2:

The results of vehicle trajectory clustering between groups with three hierarchies (upper) and five hierarchies (lower) are shown in Figure 11.

The trajectories between groups have a larger geographical range than those within a group. It can be seen from the visualization effect that the SCAIBG has achieved the overall trend of vehicles moving between groups. The green line represents all vehicle trajectories between groups, while the arrows in other colors represent the overall trend of vehicles moving at different hierarchies.

#### 4.5.3. Cluster Effect Evaluation

The silhouette coefficient [56] is used to compare and analyze the traditional DBSCAN-based method [57] with the proposed algorithm to verify the effectiveness of the proposed method. The average distance between the sample and other samples in the same cluster and the average distance between the sample and the next nearest cluster are combined for evaluation:(33)$$Silhouette\;Coefficient\left(\zeta \right)=\left|1-a\left(\zeta \right)/b\left(\zeta \right)\right|\in \left[-1,1\right].$$

Figure 12 shows the Silhouette Coefficient comparison of trajectory clustering within a group.

Figure 13 shows the silhouette coefficient comparison of trajectory clustering between groups.

For SCAIMG, the point correspondence of trajectory is considered when trajectory clustering within a group is so that the unequal-length trajectories can move reasonably according to the corresponding relation. The distance overlapping scale factor is used to offset the unreasonable distance increase between trajectories caused by camera overlap to obtain accurate trajectory centers. However, the traditional algorithm does not consider the corresponding relation between trajectories. It can only obtain the approximate distance between trajectories when measuring the trajectory similarity, so the cluster center obtained from the clustering results can not accurately represent the overall trend of trajectories. In addition, the camera number with the most capture times is used as the clustering label for supervised clustering, considering the spatial relationship between the camera and the video scenario. Figure 12 reveals that the SCAIMG is superior to the traditional trajectory clustering algorithm in clustering each camera group in large-scale vehicle trajectory data such as CityFlowV2.

For the trajectories between groups, similar to SCAIMG, SCAIBG takes group number as the cluster label for supervised clustering and considers the spatial position of camera groups and the spatial relation of video scenario, so the clustering effect is also better than that of the traditional algorithm, as shown in Figure 13.

#### 4.5.4. Algorithm Time Analysis

For SCAIMG, it is assumed that there are n video objects. It takes $O\left(n\right)$. to obtain the overlap scale factor for each trajectory. The corresponding relation and distance between each pair of trajectories can be calculated before the iteration. The corresponding relation between each pair of trajectories can be accelerated to $O\left(n\right)$ through coarse-graining, projection, and fine-graining [58], so it takes time $\leq n\cdot O\left(n\right)=O\left(n^{2}\right)$, in general,
(34)$$T_{M}\left(n\right)\leq O\left(n^{2}\right).\;$$

Therefore, the algorithm maintains linear time complexity, proving that the algorithm is effective.

Under the hardware given, the CPU time required by SCAIBG on the CityFlowV2 dataset is shown in Table 3.

The algorithm can obtain clustering results in a few seconds or even one second in groups other than groups 2 and 3. The longer time in groups 2 and 3 is because some trajectories within the group are long, taking it extra time to calculate the distance. Overall, SCAIMG can quickly extract the cluster center of large-scale vehicle trajectories.

For the SCAIBG, the clustering results in λ group of prototype vectors, with obj_num video objects. For each video object $obj\left(u\right)$, it passes through $Cnum\left(f_{obj\left(u\right)}\right)$ camera groups. Therefore, there should be u−λ−1 subsegment of the trajectory under hierarchy u. It takes $O\left(K_{t}\right)$ to randomly select $K_{t}$ samples. In general, if the iterations in the learning phase reach epochs or the cluster center does not change, it will take a time of
(35)$$T_{B}\left(n\right)\leq O\left(Iter_{max}\cdot \overset{obj_{num}}{\underset{u=1}{\sum }}Cnum\left(f_{obj\left(t\right)}\right)\cdot K_{t}\right)=O\left(n\right).$$

$Iter_{max}$, $obj_{num}$, $Cnum\left(f_{obj\left(t\right)}\right)$, and $K_{t}$ represent the maximum number of iterations, the number of video objects, the number of cross-camera trajectories corresponding to video objects, and the number of cluster centers, respectively. The linear time complexity of the algorithm proves that the algorithm is effective.

Under the hardware given, the CPU time required for different hierarchies of trajectories is shown in Table 4.

SCAIBG can effectively simplify the calculation of regular distance by extracting the start and end points of the trajectory. Therefore, it can be seen from Table 4 that the SCAIBG performs quite fast in extracting cluster centers between groups in large-scale vehicle datasets.

## 5. Discussion

The purpose of “Hierarchical Clustering Algorithm for Multi-camera Vehicle Trajectories” proposed in this paper is to represent vehicle video objects with similar motion patterns by obtaining different levels of video object clustering centers as the representative of vehicle video object motion patterns. The method in this paper has and is not limited to the following practical applications:(1)Multi-scale vehicle path prediction. According to priori theories, we can combine time information to analyze the multi-scale prediction of vehicle moving path.(2)Urban Planning and Road Planning. Different levels of clustering results, combined with road constraints, provide effective experience and technical support for urban planning departments and traffic management departments.(3)Trajectory outliers detection. The clustering results can provide a prerequisite for vehicle trajectory outliers detection. Trajectory outliers are a small number of trajectories that are obviously different from other trajectories data in the trajectory data set, so vehicle trajectory outliers can be analyzed through the distance between the vehicle trajectories and clustering centers.

In the above applications, we think that the trajectory clustering algorithm should ensure high accuracy and efficiency. High accuracy is the premise of the accuracy of the follow-up work, and efficiency is the rigid requirement of some practical applications. For example, path prediction analysis is applied to navigation planning and trajectory anomaly detection to security departments. Rapid cluster analysis enables these applications to be implemented safely.

In this paper, the elbow method is used to determine the optimal number of clusters in clustering analysis, but this method may have some shortcomings in practical application, because the vehicle trajectory has road constraints, and there is a deviation simply according to the elbow method to determine the clustering center without taking into account the actual situation.

Therefore, in the future scientific research work, the final number of clustering centers should be determined by combining the algorithm and road constraints.

## 6. Conclusion and Prospects

This paper proposed a multi-camera vehicle trajectory hierarchical clustering algorithm based on spatio-temporal grouping, aiming at the problem that the traditional trajectory clustering algorithm did not consider the camera position information, the field of view, and the hierarchical relation of the video object movement between the camera and the scenario. This method used camera number and group number as labels and was a supervised clustering algorithm considering camera information applied to trajectories within and between groups, respectively, enabling trajectories to be classified into reasonable clustering centers.

The experiment showed the clustering results and the comparison and analysis of silhouette coefficients between the proposed method and other clustering methods, and the time complexity analysis proved the proposed algorithm’s accuracy, effectiveness, and high efficiency. The limitation of this paper lies in the lack of trajectory clustering analysis combined with spatial semantic features, which will be improved in future research.

## Figures and Tables

**Figure 1 sensors-23-06909-f001:**
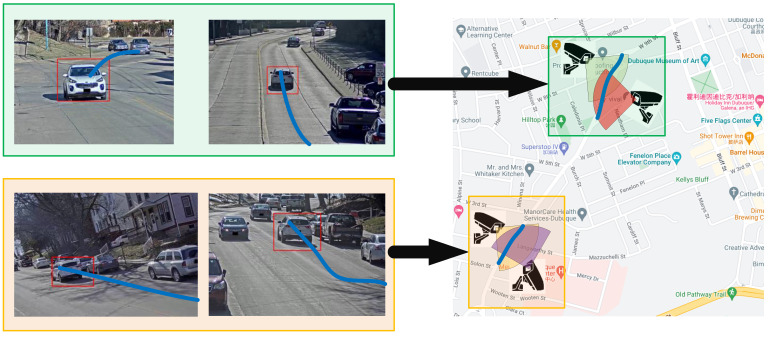
Schematic diagram of video object trajectory spatialization (blue lines represent the image space trajectories and the geospatial trajectories of the vehicle video object).

**Figure 2 sensors-23-06909-f002:**
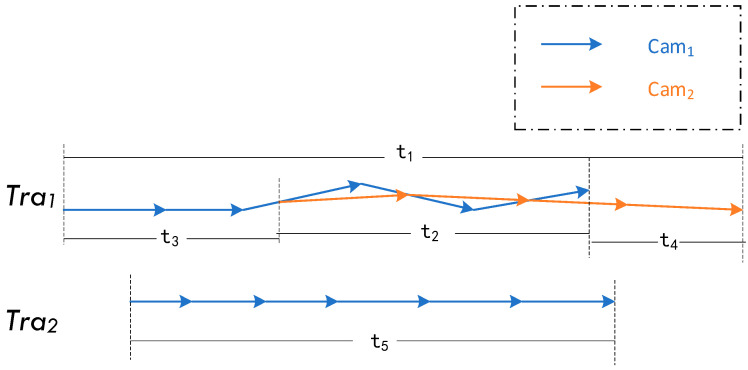
Schematic diagram of trajectory capture camera overlay.

**Figure 3 sensors-23-06909-f003:**
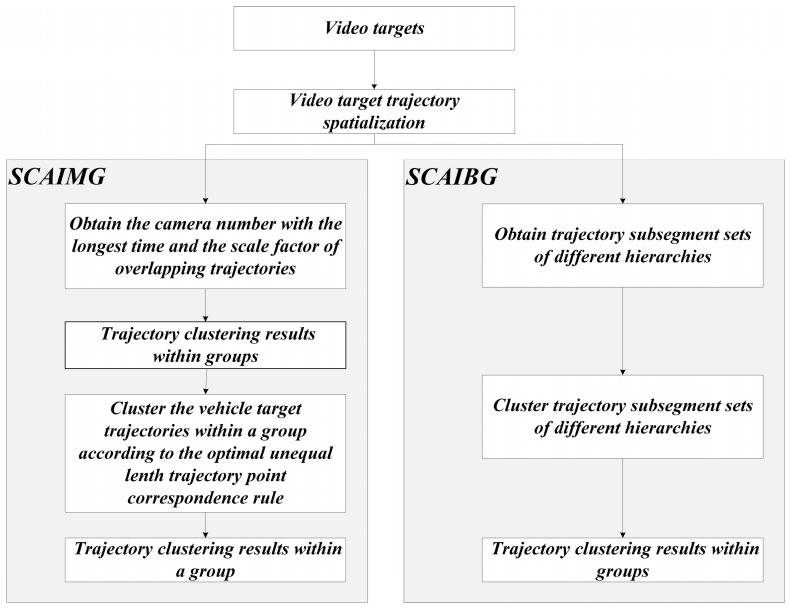
Flow chart of hierarchical clustering algorithm for vehicle trajectory considering camera information under the multi-camera joint monitoring system.

**Figure 4 sensors-23-06909-f004:**
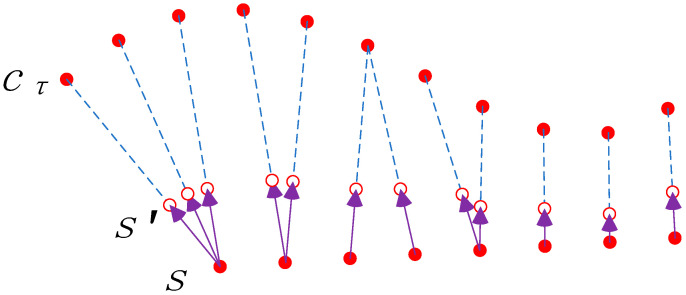
Schematic diagram of trajectory “approaching” the cluster center.

**Figure 5 sensors-23-06909-f005:**
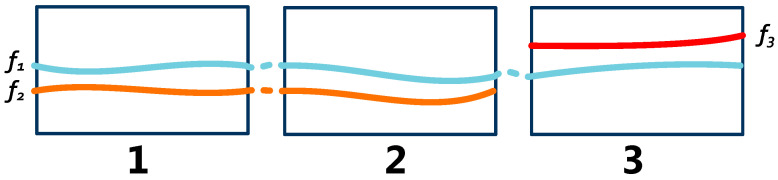
Trajectory subsegments.

**Figure 6 sensors-23-06909-f006:**
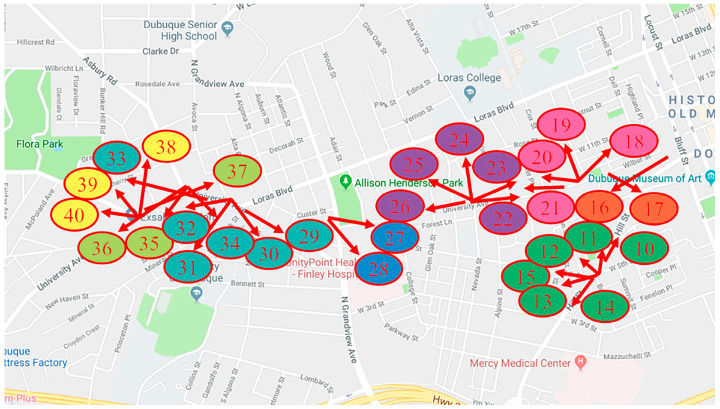
Camera Grouping Results of the CityFlowV2 Dataset (numbers represent camera numbers, red arrows represent the main optical axis direction of the camera lens, and ellipses of different colors represent different camera groups).

**Figure 7 sensors-23-06909-f007:**
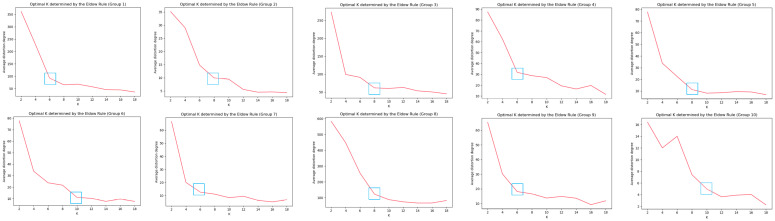
Schematic diagram for determining the number of cluster centers of camera groups in SCAIMG (blue boxes represents the optimal K value).

**Figure 8 sensors-23-06909-f008:**
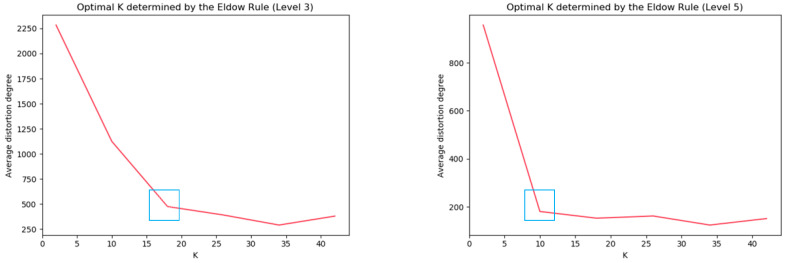
Schematic diagram for determining the number of clustering centers of three and five hierarchies in SCAIBG (blue boxes represents the optimal K value).

**Figure 9 sensors-23-06909-f009:**
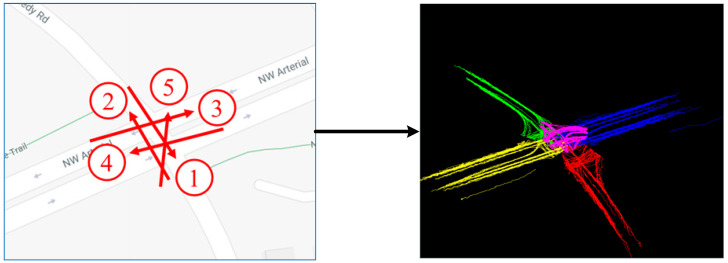
Schematic diagram of trajectory acquisition of each camera in Group 1 (numbers represent camera numbers, red arrows represent the main optical axis direction of the camera lens, and different color lines represent trajectories under different cameras).

**Figure 10 sensors-23-06909-f010:**
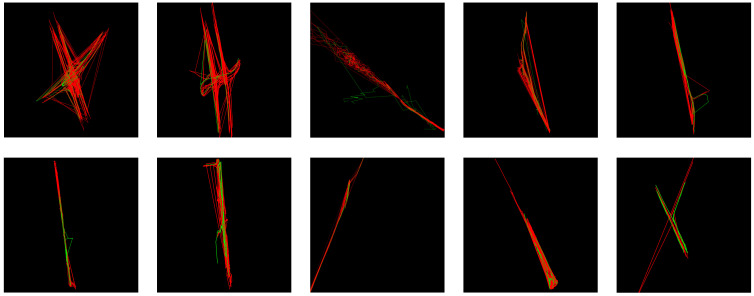
Schematic diagram of 10 groups of vehicle trajectory clustering in the CityflowV2 dataset (red lines represent the trajectory, and green lines represent the cluster center).

**Figure 11 sensors-23-06909-f011:**
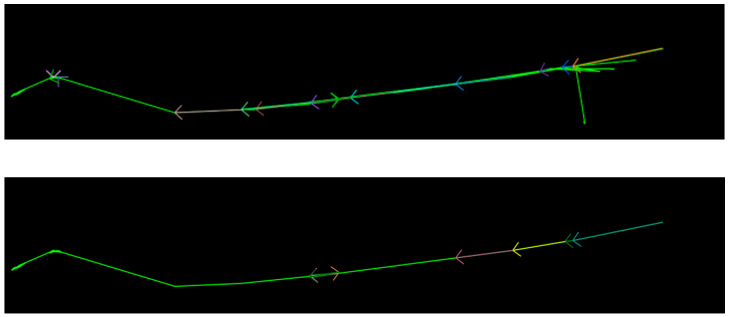
Schematic diagram of vehicle trajectory clustering between groups (three and five hierarchies; arrows of different colors represent cluster centers).

**Figure 12 sensors-23-06909-f012:**
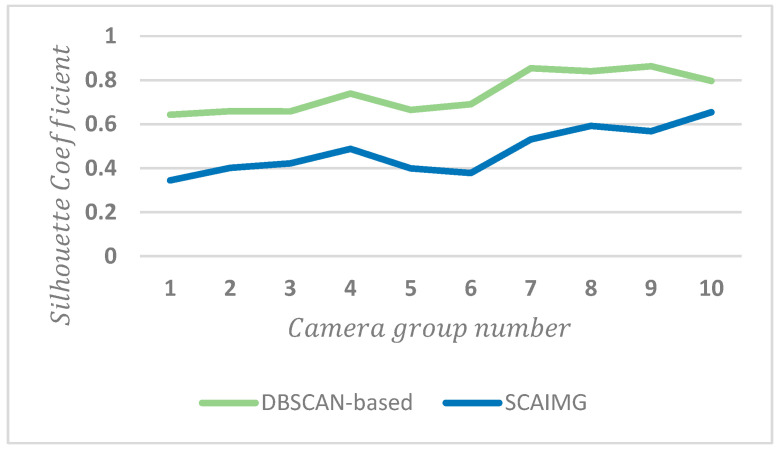
Silhouette coefficient comparison of trajectory clustering within a camera group.

**Figure 13 sensors-23-06909-f013:**
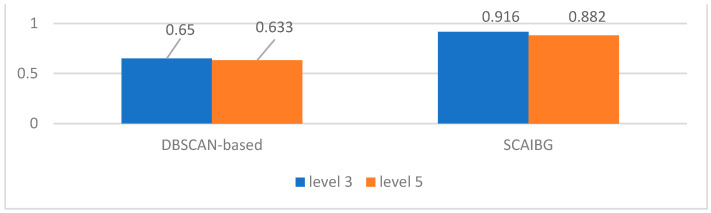
Silhouette coefficient comparison of trajectory clustering between camera groups.

**Table 1 sensors-23-06909-t001:** Tabular overview of different trajectory clustering algorithms.

Category	Classification	Time Complexity	Antinoise Ability	Labor Cost	Measurement	Representative Algorithm
Unsupervised	Densely Models	O(n*logn)	General	low	Density	DBCSAN [33]
	Hierarchical Models	O(n^2^logn)	Strong	low	Distance	HITS [36]
	Spectral Models	O(n^3^)	Strong	low	Distance	CURE [37]
Supervised	Nearest Neighbor	O(n)	weak	high	Distance	K-NN [39]
	Statistical Models	O(n^2^)	weak	high	Pattern mining	GMM [41]
	Neural Network	Depends on network	General	high	Deep network characteristics	CNN [42]
Semi-supervised	Invented from unsupervised or supervised algorithms	Related to the invented algorithm	Related to the invented algorithm	General	Related to the invented algorithm	Modified Hierarchical Clustering Models [43], modified Statistical Models [44], etc.

**Table 2 sensors-23-06909-t002:** Number of trajectory subsegments (three and five hierarchies).

Hierarchies	Number of Trajectory Subsegments
3	460
5	391

**Table 3 sensors-23-06909-t003:** The CPU time required by SCAIMG for each group in the CityFlowV2 dataset.

Hierarchies	Number of Trajectory Subsegments
1	5.693
2	10.165
3	14.770
4	6.558
5	0.388
6	1.343
7	0.359
8	0.559
9	0.465
10	2.642

**Table 4 sensors-23-06909-t004:** The CPU time required by SCAIBG for each group in the CityFlowV2 dataset.

Hierarchies	Number of Trajectories	CPU Time
3	14.770	0.007
5	0.388	0.036

## Data Availability

CityFlowV2 dataset at https://www.aicitychallenge.org (accessed on 2 July 2023).

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
