# Peer review of "Hierarchical Clustering Algorithm for Multi-Camera Vehicle Trajectories Based on Spatio-Temporal Grouping under Intelligent Transportation and Smart City"

_sensors, 2023, doi:10.3390/s23156909_

Round 1

Reviewer 1 Report

This paper represents quite solid scientific contribution on highly relevant topic. The results are promising and method well-described. Despite that most aspects are on publication-ready level, there is still space for several improvements:

 - Naming of subsections and sections should be reviewed by authors and revised slightly, as it seems a bit confusing currently. There is a whole section entitled “Basic work”, where the method itself is described. Therefore, I would recommend to rename it to something more descriptive when it comes to the content itself, such as “Approach” or “Method description” or “Implementation”. Better take a look on other papers from this journal and make it more uniform to journal format.

- Conclusion should be a distinct section, not a subsection

- Related works can be extended a bit. Moreover, tabular overview of relevant solutions, their capabilities, results and underlying datasets could be summarized 

No major issues detected. In general, the quality of language-related aspects is on acceptable level. However, improvements are also recommended here. Overall, there are many sentences which seem quite long and make the text more difficult to follow. Therefore, it is recommended to revise them and split into several shorter formulations.

Reviewer 2 Report

MDPI Sensors Journal (Manuscript ID: sensors- 2475019)

Comments to the Author

This paper proposes a hierarchical clustering algorithm for multi-camera vehicle trajectories based on spatio-temporal grouping. The paper touches on an interesting topic. However, there are several points that need to be addressed to improve the quality of the manuscript.

Suggestions to improve the quality of the paper are provided below:

1)     Given that this work is an extension of previous work, please clearly state the contributions of this work and the ways in which it improves upon the previous work.

2)     The current literature review is very brief and should be further extended by discussing in greater details different studies that have adopted supervised, semi-supervised, and unsupervised approaches for trajectory clustering. There should be sufficient details provided to (1) help readers have a working understanding of the existing state-of-art, and (2) identify the novelty of the proposed work.

3)     Given that the initial cluster centres for SCAIMG and SCAIBG are randomly selected, this usually results in slightly different clusters at the end when the clustering algorithm is rerun-ed. Some studies that have adopted a semi-supervised clustering approach have proposed different ways to deal with this issue, such as initialising the cluster centres using labelled data [https://doi.org/10.1016/j.buildenv.2020.106681]. In the manuscript, please elaborate on this issue by comparing the approach in the suggested work, and briefly discuss how this issue is resolved in this paper.

4)     Please include a discussion section to talk about the implications of this work, as well as the limitations of this work and how it will be addressed in future works. For instance:

·       What are the potential benefits of obtaining these clusters in a short amount of time?

·       What are the applications of this work?

·       Given that the optimal number of clusters is determined using the elbow method, how can this process be automated in the future using a more objective approach and without human input?

5)     Minor comments on paper structure and writing

·       The first paragraph of Section 3 seems to be template text. Please check the manuscript thoroughly and remove text that is not related to the work.

·       Remove the numbers on the line graphs for Figure 12 to reduce the clutter on the figure.

·       Replace Figure 13 with a 2D diagram instead of a 3D diagram.

The level of English is satisfactory.

Round 2

Reviewer 2 Report

Thank you for addressing my comments and concerns carefully. This paper is ready for publication.